# Evaluation of Antibacterial and Antiviral Compounds from *Commiphora myrrha* (T.Nees) Engl. Resin and Their Promising Application with Biochar

**Jin Woo Kim [1,2], Saerom Park [1,3], Young Whan Sung [1], Hak Jin Song [1], Sung Woo Yang [1], Jiwoo Han [1], Jeong Wook Jo [1], Im-Soon Lee [4], Sang Hyun Lee [1], Yong-Keun Choi [1,3,*] and Hyung Joo Kim [1,*]**

[1] Department of Biological Engineering, Konkuk University, Seoul 05029, Republic of Korea; healthok9988@naver.com (J.W.K.); angel4y@naver.com (S.P.); lamiavita@naver.com (Y.W.S.); hjeda11@naver.com (H.J.S.); rick98@naver.com (S.W.Y.); jw991021@naver.com (J.H.); jjw9802@naver.com (J.W.J.); sanghlee@konkuk.ac.kr (S.H.L.)

[2] R&D Center, Myrrhzone Molyac Institute Co., Ltd., Jincheon 27856, Chungcheongbuk-do, Republic of Korea

[3] R&D Center, Choilab Inc., Seoul 01811, Republic of Korea

[4] Department of Biological Sciences, Konkuk University, Seoul 05029, Republic of Korea; islee@konkuk.ac.kr

[*] Correspondence: dragonrt@konkuk.ac.kr (Y.-K.C.); hyungkim@konkuk.ac.kr (H.J.K.); Tel.: +82-02-2049-6111 (Y.-K.C. & H.J.K.); Fax: +82-02-3437-8360 (Y.-K.C. & H.J.K.)

**Abstract:** *Commiphora myrrha* (T.Nees) Engl. resin extracts were prepared via immersion in extraction solvents (hot water, DMSO, hexane, ethanol, and methanol) which have various physical properties, such as different polarity and dielectric constant values. Methanolic *C. myrrha* (T.Nees) Engl. resin extracts showed broad antibacterial activity against isolated airborne bacteria. All methanolic *C. myrrha* (T.Nees) Engl. resin extracts were analyzed using GC-MS and Furanoeudesma-1,3-diene and curzerene were found as the main terpenoids. In addition, the methanolic *C. myrrha* (T.Nees) Engl. resin extracts were found to have antiviral activity (81.2% viral RNA inhibition) against the H1N1 influenza virus. Biochars (wood powder- and rice husk-derived) coated with *C. myrrha* (T.Nees) Engl. resin extracts also showed antiviral activity (22.6% and 24.3% viral RNA inhibition) due to the adsorption of terpenoids onto biochar. *C. myrrha* (T.Nees) Engl. resin extract using methanol as the extraction solvent is a promising agent with antibacterial and antiviral efficacy that can be utilized as a novel material via adsorption onto biochar for air filtration processes, cosmetics, fertilizers, drug delivery, and corrosion inhibition.

**Keywords:** *C. myrrha* (T.Nees) Engl.; antivirus; furanoeudesma-1,3-diene; curzerene; biochar

## 1. Introduction

Naturally derived compounds extracted from natural sources (e.g., plants, marine organisms, microorganisms, and animals) are of growing interest because of their non-toxic properties and high biocompatibility compared with chemically synthesized compounds (e.g., β-lactams such as penicillin and cephalosporins) [1,2]. Natural compounds, including flavonoids, phenolic compounds, terpenoids, etc., are known to have various biological activities (e.g., antimicrobial, antiviral, and so on) [3–7]. Several researchers have studied these natural compounds to analyze their physicochemical (e.g., terpenes, terpenoids, and secondary metabolites) and biological properties (e.g., antimicrobial activity against bacteria, fungi, and viruses) [3–8].

In recent years, microorganisms such as viruses and bacteria have become a crucial issue in both animal diseases and life-threatening diseases affecting humans (e.g., COVID-19). Humans are routinely exposed to airborne pathogenic microorganisms; therefore, prevention strategies are necessary. According to studies, extracts derived from natural sources (e.g., plants, marine origin, etc.) exhibit a resistance to microorganisms [9–13]. Thus, naturally derived compounds may be potential materials for solving microbial threats.

In previous studies, supercritical $CO_2$, enzyme-, ultrasonic-, and microwave-assisted extractions have been introduced as alternatives to conventional solvent extraction processes for more effective extraction of bioactive molecules from natural materials [8,14–16]. Although these extraction techniques have some advantages, they are limited by various factors (e.g., high cost, complicated processes, etc.) [17]. Consequently, numerous studies have employed the solvent extraction method to acquire naturally derived compounds. Investigating the most optimal extraction solvent is necessary because variations in the performance of the extraction solvents and raw materials (e.g., plants) lead to differences in outcomes [18–24]. Similarly, several authors have reported that the amount of extracted polyphenols and the antibacterial ability of the extracts are altered according to the polarity of the extraction solvent [11,22,24].

*C. myrrha* resin produced from the *Commiphora* genus (mostly *C. myrrha* and *C. molmol* of 150 species in Africa, Arabia, and India) has been used for various therapeutic applications (e.g., embalming ointment, pain reliever, etc.) due to the prevention and treatment performance of several components, such as terpenes, steroids, and sterols [16,23]. Therefore, *C. myrrha* resin is beneficial as a medicinal agent for infection prevention and wound treatment [25]. In addition, *C. myrrha* resin shows antimicrobial activity against bacteria such as *Staphylococcus aureus*, *Bacillus cereus*, *Bacillus subtilis*, *Escherichia coli*, *Klebsiella pneumoniae*, and *Fusobacterium nucleatum* [26–30]. Madia et al., analyzed the antiviral activity of *C. myrrha* extracts obtained using a supercritical fluid extraction process against influenza A virus [16]. Extracts containing alpha-tocopherol acetate (ATA) decrease viral replication and nucleoprotein expression. However, a few studies have investigated the viral resistance of *C. myrrha* resins.

Researchers have attempted to study antimicrobial materials and synthesize them using phytochemicals for potential applications. Modern technologies include bionanoparticles (for therapy and drug delivery), Ag nanoparticle composites (for antimicrobial and antiviral purposes), and biochar (for environmental remediation) [31–33]. Strasakova et al., reported that caraway essential oils, including aromatic compounds such as terpinene, cymene, and limonene, need to be immobilized and incorporated into a matrix (e.g., polypropylene) owing to their inherent volatility [34]. Overcoming these problems can reduce the loss of compounds and maintain desired compounds. Therefore, biochar has attracted considerable attention because of its advantages such as cost-effectiveness, eco-friendliness (e.g., usage of waste biomass and $CO_2$ storage), adsorption potential, and easy functionalization compared with other carbonaceous materials (e.g., activated carbon and graphene) [35,36]. Currently, biochar produced from various biomass wastes (e.g., wood, grass, and so on) under $O_2$ limitation has been used as a soil amendment and adsorbent for remediation [37–41]. However, the utilization of biochar has been steadily increasing across various fields (e.g., cosmetics, fertilizers, etc.) in recent years, with the potential for further expansion. However, to the best of our knowledge, the potential of biochar coated with naturally derived compounds as an antiviral agent has rarely been evaluated. Thus, the *C. myrrha* (T.Nees) Engl. resin extract-coated biochar is a promising material with antiviral and antibacterial activities.

Therefore, this study focused on the evaluation of the antibacterial activity of *C. myrrha* (T.Nees) Engl. resin extracts against isolated airborne bacteria and the antiviral activity against the H1N1 influenza virus. First, the optimal extraction solvent for *C. myrrha* (T.Nees) Engl. resin extracts were investigated through an antibacterial activity test using the disk diffusion method. Second, antiviral activity, cytotoxicity, and anti-inflammatory tests were conducted using the *C. myrrha* (T.Nees) Engl. resin extracts with the chosen optimal extraction solvent. Furthermore, this study assessed the properties of *C. myrrha* (T.Nees) Engl. resin extract-coated biochar (e.g., surface changes of biochar using FTIR) as a promising application and identified possible compounds (e.g., terpenoids) with antiviral activity.

## 2. Materials and Methods

### 2.1. Materials

*Commiphora myrrha* (T.Nees) Engl. resin was obtained from the KT&I Trade Industry (*C. myrrha* Gum, Addis Ababa, Ethiopia) and powdered using a mortar. Pure water (HPLC grade), ethanol (99%), methanol (99%), and DMSO (dimethyl sulfoxide; 99%) as solvents for extraction were purchased from Sigma-Aldrich (Seoul, Republic of Korea). In addition, *Achyranthes bidentata* Blume root was extracted using methanol and was used for antiviral activity as the control experiment. Tryptic soy agar, tryptic soy broth, and nutrient broth used as media for bacterial cultures were purchased from Difco (Seoul, Republic of Korea). Rice husk and wood powder were used as waste biomass for biochar production. The rice husk and wood powders were oven-dried at 100 °C for 24 h and ground to particle sizes ranging from 100 to 200 μm. For biochar production, pyrolysis of rice husk and wood powder was conducted at the temperature of 550 °C for 2 h under $N_2$ gas. Rice husk-derived biochar (RH-BC) and wood powder-derived biochar (WD-BC) were washed several times with distilled water to remove impurities.

### 2.2. Preparation of C. myrrha (T.Nees) Engl. Resin Extracts

Dried *C. myrrha* (T.Nees) Engl. resin (2.5 g) was weighed into a vial, and 10 mL of solvents (e.g., pure water, ethanol, methanol, and DMSO) were added for extraction. For the hot water extraction, the mixtures of *C. myrrha* (T.Nees) Engl. resin powder and pure water were shaken at 180 rpm for 3 h at 80 °C. For the extraction of *C. myrrha* (T.Nees) Engl. resin using ethanol, methanol, and DMSO, mixtures of *C. myrrha* (T.Nees) Engl. resin powder and each solvent were vigorously shaken at 180 rpm for 24 h at room temperature. Each solution was centrifuged at 3500 rpm and filtered through a 0.2 μm PVDF syringe filter. The vials containing the extracts were then stored at 4 °C for subsequent experiments.

### 2.3. Isolation and Identification of Airborne Bacterial Strains

Airborne bacterial strains were isolated for the antibacterial experiments. The airborne bacterial strains were collected and cultured on tryptic soy agar plates. Airborne bacterial strains were selected based on differences in colony morphology and color. All bacterial strains were subcultured on nutrient agar for identification at 30 °C for 24 h. They were cultured in tryptic soy broth for the antibacterial experiments at 30 °C for 24 h. The isolated bacterial strains were identified using 16S rRNA sequencing by Macrogen (Seoul, Republic of Korea). Two oligonucleotide primers (forward:27F and reverse:1492R) were used as universal prokaryotic primers for amplifying the bacterial 16S rRNA gene. The isolated bacterial strains were identified using the BLAST Basic Local Alignment Search Tool (BLAST).

### 2.4. Antibacterial Activity Evaluation of C. myrrha (T.Nees) Engl. Resin Extracts

The antibacterial activity of *C. myrrha* (T.Nees) Engl. resin extracts was evaluated using the disk diffusion method described by Kang et al. [42]. For the antibacterial activity evaluation, the isolated strains were cultivated in sterile tryptic soy broth at 30 °C for 24 h. Cultures were spread on the surfaces of tryptic soy agar plates. The *C. myrrha* (T.Nees) Engl. resin extract was loaded into disks and the solvents were evaporated, except for the DMSO. Then, the disks were placed on agar plates. Also, all solvents were evaluated as a control experiment. All plates were incubated at 30 °C for 24 h and the inhibition zone diameter was measured.

### 2.5. Evaluation of Cytotoxicity and Anti-Inflammatory Properties of C. myrrha (T.Nees) Engl. Resin Extracts

For in vitro cytotoxicity of *C. myrrha* (T.Nees) Engl. resin extracts, cells from the murine macrophage-like cell line RAW 264.7 were obtained from the Korean Cell Line Bank (Seoul, Republic of Korea). The cells were cultured in Dulbecco's modified Eagle's medium (DMEM; WelGENE Inc., Daegu, Republic of Korea) supplemented with 10% fetal

bovine serum (FBS; Gibco Laboratories, Grand Island, NY, USA) and antibiotics. Cell suspensions were seeded in 96-well flat-bottomed plates with 200 μL per well to culture $3 \times 10^4$ cells per well for 24 h. The cells were incubated with *C. myrrha* (T.Nees) Engl. resin extract at a concentration of 0.08–10% (*v/v*) for 48 h at 37 °C in a CO$_2$ incubator. Cell viability was measured using the NR assay (Neutral Red). The cells were treated with a Neutral Red (NR; Sigma Aldrich Corporation, Saint Louis, MO, USA) solution containing DMEM for 3 h to dye the lysosomes of the living cells. After exposure to the NR solution, the supernatant was completely removed and treated with a desorption solution to extract the dye from the cells. The desorbed solution was made with a 49% ethanol solution (*v/v*) containing 1% acetic acid (*v/v*). Absorbance was measured at 540 nm using a microplate spectrophotometer (μ2 Micro Digital, MOBI, Seoul, Republic of Korea).

The anti-inflammatory activity of the *C. myrrha* (T.Nees) Engl. resin extracts was evaluated by measuring the amount of nitric oxide produced by Raw 264.7 cells within the ranges of low cytotoxicity. RAW 264.7 cell suspensions were seeded in 96-well flat-bottomed plates with 200 μL per well to culture $3 \times 10^4$ cells per well for 24 h. The cells were incubated with *C. myrrha* (T.Nees) Engl. resin extract at a concentration of 0.08–10% (*v/v*) for 48 h at 37 °C in a CO$_2$ incubator. The *C. myrrha* (T.Nees) Engl. resin extract was diluted with DMEM media with no phenol red, supplemented with 10% fetal bovine serum, antibiotics, and 1 μg/mL lipopolysaccharide from *Escherichia coli* (LPS; Sigma Aldrich Corporation, Saint Louis, MO, USA). Nitrite release in the culture medium was determined by transferring the supernatant (100 μL) to a new 96-well flat-bottomed plate and adding 100 μL of the Griess reaction to each well. The plates were incubated at room temperature for 15 min. Absorbance was measured at 540 nm using a microplate spectrophotometer (MOBI). NO levels were estimated to assess the anti-inflammatory effect of *C. myrrha* (T.Nees) Engl. resin extract by determining the decrease in NO concentration in the media that was provoked by the LPS.

### 2.6. Antiviral Activity of C. myrrha (T.Nees) Engl. Resin Extracts and C. myrrha (T.Nees) Engl. Resin Extract-Coated Biochar

The H1N1 influenza virus (Influenza A/Human/Korea/KUMC-33/2005, obtained from the Korea Bank for Pathogenic Viruses (Seoul, Republic of Korea)) used in this study was tested for antiviral activity. First, the antiviral activity of methanol as a control and *C. myrrha* (T.Nees) Engl. resin extract was evaluated against the H1N1 influenza virus. The concentration of H1N1 influenza virus was adjusted to 0.1 of MOI (Multiplicity of infection; $5.0 \times 10^7$ plaque-forming units (PFU)/mL)) in 1.5 mL of aqueous solution with 100 μL of methanol and *C. myrrha* (T.Nees) Engl. resin extracts in methanol. In addition, concentrations of the standard samples (e.g., furanoeudesma-1,3-diene and curzerene) ranging from 10–90 μL/mL were investigated for their antiviral activity. To evaluate the antiviral activity of *C. myrrha* (T.Nees) Engl. resin extract-coated biochar, 5 mg of *C. myrrha* (T.Nees) Engl. resin extract-coated biochar was added to a vial containing water (3 mL) with virus ($5.0 \times 10^7$ plaque-forming units (PFU)/mL). The mixture was shaken at 120 rpm and stored at 25 °C for 48 h. Subsequently, the mixture was centrifuged at 13,000 rpm for 5 min and filtrated through a 0.2 μm PVDF syringe filter. After the reaction, the residual virus concentration and viral RNA inhibition were measured using qRT-PCR after RNA extraction from the harvested mixture solution [43]. The residual virus concentration and viral RNA inhibition were compared with those of biochar without a *C. myrrha* (T.Nees) Engl. resin extract coating. In addition, a plaque assay was conducted according to the previous study [43].

### 2.7. Adsorption of C. myrrha (T.Nees) Engl. Resin Extracts onto Biochar

To adsorb the *C. myrrha* (T.Nees) Engl. resin extracts onto biochar (RH-BC and WD-BC), 50 mg of biochar was mixed with 1 mL of the extract solution in a vial. The mixtures were then placed in a shaking incubator at 25 °C for 24 h at 120 rpm. The *C. myrrha* (T.Nees) Engl. resin extract-coated biochar and the residue solution were separated via centrifugation at

3500 rpm for 20 min. The separated biochar was dried at 60 °C for 24 h to remove methanol. The functional groups of the separated biochars were investigated using Fourier-transform infrared (FTIR) spectroscopy (FT/IR-4600 spectrometer; Jasco, Tokyo, Japan). The FTIR spectra were subsequently compared with those of the biochar without extract adsorption. The changes in polyphenols and terpenoids in the initial (with extract adsorption) and final solutions (without extract adsorption) were determined using HPLC and GC-MS.

To identify the thermal stability of various compounds in *C. myrrha* (T.Nees) Engl. resin extracts on the surface of biochar, *C. myrrha* (T.Nees) Engl. resin extract-coated RH-BC were neglected in oven under different temperatures (25 °C, 50 °C, 100 °C, and 200 °C) for 1 h. Changes in the surface of the biochar were observed using X-ray photoelectron spectroscopy (XPS, K-Alpha, Thermo Scientific, Waltham, MA, USA).

### 2.8. Natural Compound Analysis of C. myrrha (T.Nees) Engl. Resin Extracts

Various natural compounds in the *C. myrrha* (T.Nees) Engl. resin extract were analyzed using HPLC and GC-MS for polyphenols and terpenoids. Centrifugation and filtration were performed to separate soluble compounds and insoluble materials. In addition, the supernatant of the *C. myrrha* (T.Nees) Engl. resin extract stored at 4 °C was used for analysis. Polyphenols were detected using HPLC (YL 9100 system, Younglin, Republic of Korea) with a UV detector. The chromatograms were determined at 254 nm with YMC-Triart C18 column (250 mm × 4.6 × 5 µm) (YMC, Seoul, Republic of Korea) at 25 °C under the gradient condition of mobile phase A (4% acetic acid in water) and mobile phase B (Methanol) with the flow rate of 0.5 mL/min. The gradient program was initiated with 100% of A solution and was held for the first 5 min. The concentration of A solution was followed by 50% of eluent B for the next 7 min. This was followed by 80% of eluent B for the next 10 min. Subsequently, this percentage was changed to 50% and 100% for 6 min and 7 min. Finally, the mixtures were incubated for 5 min. The polyphenols in the *C. myrrha* (T.Nees) Engl. resin extracts were compared with authentic polyphenols (Figure S1). The detection of terpenoids was investigated using GC-MS (Perkin Elmer, Waltham, MA, USA) with a fused silica capillary column (Elite-5 ms, 30 m × 0.25 mm i.d. × 0.25 µm). The analytical conditions were as follows: 250 °C (inlet temperature), increased from 40 °C for 1 min to 120 °C for 2 min, followed by 15 °C for 1 min, and then increased to 3000 °C for 5 min at 10 °C/min.

### 2.9. Statistical Analysis

The one-way ANOVA with Tukey's HSD test was conducted using Minitab 16.0 software (Minitab Inc., State College, PA, USA). The statistical significance was set as $p < 0.05$.

## 3. Results and Discussion

### 3.1. Identification of Test Microorganisms

To obtain bacterial strains for analysis of the antibacterial activity of the *C. myrrha* (T.Nees) Engl. resin extracts, airborne bacteria were collected and isolated. The isolated bacterial strains showed high homology to *Paenibacillus pasadenensis* NBRC 161214 (98% homology), *Micrococcus yunnanensis* YIM 65004 (99% homology), *Pseudemonas azotoformans* NBRC 12693 (99% homology), *Rhodococcus qingshengii* dj1-6-2 (99% homology), *Staphylococcus capitis* JCM 2420 (99% homology), *Staphylococcus epidermidis* NBRC 100911 (98% homology), and *Deinococcus radiodurans* R1 (99% homology). The isolated strains were mostly Gram-positive, except for *Pseudemonas azotoformans* NBRC12693. These strains, which are found in soil, water, sewage, and human skin, have various characteristics (e.g., pollutant degradation, infection of cereal grains, carbendazim degradation, and radiation resistance) (Table 1) [44–49]. Hence, these airborne bacterial strains could be used as potential samples for testing antibacterial activity.

**Table 1.** Isolated airborne bacteria and their characteristics. Conditions: The isolated strains were cultivated in sterile tryptic soy agar plates at 30 °C for 24 h.

| Name of Strain | Homology (%) | Morphology | Gram Staining | Remark | Ref. |
|---|---|---|---|---|---|
| *Paenibacillus pasadenensis* NBRC 161214 | 98 | Rods | Positive | Soil, water, sewage, etc. | [47] |
| *Micrococcus yunnanensis* YIM 65004 | 99 | Cocci | Positive | Pollutants degradation | [49] |
| *Pseudomonas azotoformans* NBRC 12693 | 99 | Rods | Negative | Infection of cereal grains | [45] |
| *Rhodococcus qingshengii* dj1-6-2 | 99 | Ovoid | Positive | Carbendazim degradation | [44] |
| *Staphylococcus capitis* JCM 2420 | 99 | Cocci | Positive | Human skin | [46] |
| *Staphylococcus epidermidis* NBRC 100911 | 98 | Cocci | Positive | Human skin | [46] |
| *Deinococcus radiodurans* R1 | 99 | Cocci | Positive | Radiation resistance | [48] |

*3.2. Screening of Optimal Solvent for Antibacterial Activity of C. myrrha (T.Nees) Engl. Resin Extracts on Airborne Bacterium*

The different extraction yields, antioxidant activities, and antibacterial activities depend on the type of solvent because the characteristics of the extraction solvent influence the sorting and activity of the compounds extracted from the raw materials. Extracts of *C. papaya*, *A. dealbata*, *O. europaea*, and *P. betle* Linn. using various solvents, such as ethanol, methanol, hexane, and ethyl acetate, showed different antibacterial activities depending on the physical properties of the extraction solvent [18,19,22].

Thus, the antibacterial effects of *C. myrrha* (T.Nees) Engl. resin extracts using various solvents (hot water, hexane, DMSO, methanol, and ethanol) were evaluated to determine the optimal solvent for acquiring the extract with effective antibacterial properties. The extracts were tested against isolated airborne bacteria assayed in agar plates using the disk diffusion method. *C. myrrha* (T.Nees) Engl. resin extracted using methanol and ethanol exerted antibacterial effects against *Paenibacillus pasadenensis* NBRC 161214, *Micrococcus yunnanensis* YIM 65004, and *Deinococcus radiodurans* R1 (Table 2). Additionally, an antibacterial effect of *C. myrrha* resin extracts obtained using only methanol was observed against *Rhodococcus qingshengii* dj1-6-2. However, no antibacterial effects were observed in *C. myrrha* (T.Nees) Engl. resin extracted using hot water, DMSO (except *Micrococcus yunnanensis* YIM65004), and hexane. Hence, the various extraction solvents used in obtaining *C. myrrha* (T.Nees) Engl. resin extracts are likely to influence their antibacterial effects due to differences in their effective components. In general, the antibacterial effect may be related to the cell wall structure. However, no correlation between the antibacterial effect and the cell wall structure (e.g., Gram-positive and Gram-negative strains) was observed in this study. These phenomena may be due to the action and resistance mechanisms (e.g., cell wall synthesis, nucleic acid synthesis, protein synthesis, and folic acid metabolism) of the effective components in *C. myrrha* (T.Nees) Engl. resin extracts [50]. Therefore, selecting an effective extraction solvent is an important parameter.

**Table 2.** Screening of optimal solvents for antibacterial activity of *C. myrrha* (T.Nees) Engl. resin extracts on isolated airborne bacteria. Conditions: The isolated strains were cultivated in sterile tryptic soy agar plates at 30 °C for 24 h.

| No. | Name of Strain | Antibacterial Activity | | | | |
|---|---|---|---|---|---|---|
| | | Hot Water | DMSO | Hexane | Ethanol | Methanol |
| 1 | *Paenibacillus pasadenensis* NBRC 161214 | - | - | - | + | + |
| 2 | *Micrococcus yunnanensis* YIM 65004 | - | + | - | + | + |
| 3 | *Pseudomonas azotoformans* NBRC 12693 | - | - | - | - | - |
| 4 | *Rhodococcus qingshengii* dj1-6-2 | - | - | - | - | + |
| 5 | *Staphylococcus capitis* JCM 2420 | - | - | - | - | - |
| 6 | *Staphylococcus epidermidis* NBRC 100911 | - | - | - | - | - |
| 7 | *Deinococcus radiodurans* R1 | - | - | - | + | + |

+: Positive effect; -: negative effect.

The polarity indices of the solvents used for extraction were in the following order: water (10.2) > DMSO (7.2) > ethanol (5.2) > methanol (5.1) > hexane (0.1), whereas the dielectric constants of the solvents were in the following order: water (78.54) > DMSO (47) >

methanol (32.6) > ethanol (24.3) > hexane (1.89). The polarity index and dielectric constant of solvents are associated with the extraction of hydrophobic or hydrophilic compounds from raw materials (e.g., plants and fruits). Adam and Selim and Al-Madi et al., measured the antibacterial efficiency of a *C. myrrha* (T.Nees) Engl. resin extract using ethanol and methanol as extraction solvents, respectively, and found that only the ethanolic extract exhibited antibacterial activity against *Enterococcus faecalis* [26,51]. Moreover, Mohamed et al., and Abdallah et al., also reported that the antibacterial effect of *C. myrrha* resin extract was dependent on the polarity of the extraction solvent [30,52,53]. These results may be due to differences in solvent properties (polarity and dielectric constant) that affect the type of compound being extracted. When *C. myrrha* (T.Nees) Engl. resin was extracted using hot water and ethanol, the quantity of extracted effective compounds containing carbohydrates, proteins, lipids, polyphenols, and alkaloids changed, which affected the antibacterial activity [29,53]. These results imply that the quantity of some compounds (e.g., tannic acid, rutin, and quercetin as polyphenols) in the *C. myrrha* (T.Nees) Engl. resin extracts obtained using methanol may be higher than those obtained using other solvents. Therefore, in the following experiments, methanol was used as the extraction solvent for *C. myrrha* (T.Nees) Engl. resin because it yielded active compounds with the most potent antibacterial activity against the isolated airborne bacteria. Various polyphenols including tannic acid, rutin, and quercetin were observed as three major polyphenols in the *C. myrrha* (T.Nees) Engl. resin extracts treated with methanol (Figure 1). Mandal et al., reported that tannic acid, rutin, and quercetin are effective against *Staphylococcus epidermidis* and *Pseudomonas aeruginosa* [54]. In particular, tannic acid, a major compound of methanolic *C. myrrha* (T.Nees) Engl. resin extract, inhibits the adhesion of bacteria to the surface of cells, thereby preventing microbial infection and hindering bacterial growth by reducing nutrient uptake [55,56]. Previous studies reported that tannic acid extracted from green tea, *Anthemis praecox* Link, *Quercus infectoria* galls, and *Neolamarckia cadamba* fruits showed antibacterial activity against *Staphylococcus aureus*, *Escherichia coli*, *Streptococcus pyogenes*, *Enterococcus faecalis*, *Pseudomonas aeruginosa*, *Yersinia enterocolitica*, *Listeria innocua*, and *Bacillus cereus*. Moreover, it is more sensitive to Gram-positive bacteria compared with Gram-negative bacteria [1,55–58]. These results are consistent with our study. Based on the results of previous studies, three polyphenols (e.g., tannic acid, rutin, and quercetin) are potential antibacterial substances against the airborne bacteria examined in the present study.

### 3.3. Evaluation of Antibacterial Activity of C. myrrha (T.Nees) Engl. Resin Extracts

Methanol was chosen as the optimal extraction solvent in the preceding experiments. The *C. myrrha* (T.Nees) Engl. resin extracted in methanol exhibited antibacterial activity against Gram-positive bacteria, including *Paenibacillus pasadenensis* NBRC 161214, *Micrococcus yunnanensis* YIM 65004, *Rhodococcus qingshengii* dj1-6-2, and *Deinococcus radiodurans* R1, with inhibition zones measuring 14, 10, 10, and 12 mm, respectively (Table 3). Although *Staphylococcus capitis* JCM 2420 and *Staphylococcus epidermidis* NBRC 100911 are Gram-positive bacteria, the *C. myrrha* (T.Nees) Engl. resin extract did not show any antibacterial effects against them, and *Pseudemonas azotoformans* NBRC 12693, a Gram-negative bacterium, showed resistance to the *C. myrrha* (T.Nees) Engl. resin extract. This is because the bacteria may possess different cell walls, chemically and structurally, depending on the strain (e.g., peptidoglycan layer, membrane composition, lipopolysaccharides, etc.) [59].

### 3.4. Evaluation of Cytotoxicity and Anti-Inflammatory Properties of C. myrrha (T.Nees) Engl. Resin Extracts

To evaluate the potential of *C. myrrha* (T.Nees) Engl. resin extracts as commercial products, the cytotoxicity and anti-inflammatory activity of the methanolic extracts were measured using the RAW 264.7 cell line. The cell viability significantly decreased with the addition of *C. myrrha* (T.Nees) Engl. resin extracts exceeding more than 3.75 *v/v*% (Figure 2A). The EC50 value for *C. myrrha* (T.Nees) Engl. resin extracts was measured at 4.5 *v/v*%. Therefore, we focused on the extracts within a non-cytotoxic concentra-

tion range for further investigation of their anti-inflammatory effect on RAW 264.7 cells by monitoring NO formation. In comparison to the control, when 0.08% and 2.5% of *C. myrrha* (T.Nees) Engl. resin extracts were added, the produced NO was 83.5% and 17.6%, respectively (Figure 2B). The NO yield decreased with increasing extract content. These results indicate that the *C. myrrha* (T.Nees) Engl. resin extracted using methanol has anti-inflammatory properties in vitro. Cheng et al., reported the anti-inflammatory effects of a *C. myrrha* (T.Nees) Engl. resin extract prepared by immersing in methanol on RAW 264.7 macrophages. The extract inhibited NO synthase and cyclooxygenase-2 induction, leading to reduced production of NO, prostaglandin $E_2$, interleukin-1beta, and tumor necrosis factor alpha [60].

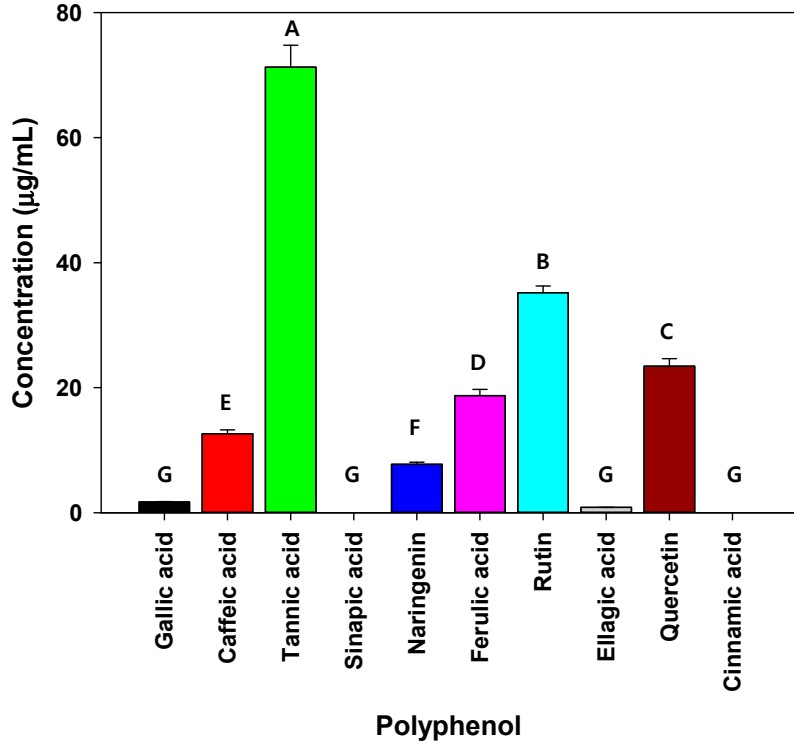

**Figure 1.** Polyphenol concentrations in *C. myrrha* (T.Nees) Engl. resin extracts analyzed using HPLC. Conditions: 254 nm; YMC-Triart C18 column; under gradient condition. The experiment was conducted in triplicate and the error bars represent the 95% confidence interval. A, B, C, D, E, F, and G indicate group classified from Tukey's test. The same letter means that there is no significant difference between the data.

**Table 3.** The antibacterial activity of *C. myrrha* (T.Nees) Engl. resin extracts using methanol against airborne bacteria. Conditions: The isolated airborne bacteria were cultivated in sterile tryptic soy agar plates at 30 °C for 24 h with disks loaded with *C. myrrha* (T.Nees) Engl. resin extracted using methanol. The experiment was conducted in triplicate and the error bars represent the 95% confidence interval.

| No. | Name of Strain | Inhibition Zone Diameter (mm) |
| --- | --- | --- |
| 1 | *Paenibacillus pasadenensis* NBRC 161214 | 14 ± 2.8 |
| 2 | *Micrococcus yunnanensis* YIM 65004 | 10 ± 0.0 |
| 3 | *Pseudomonas azotoformans* NBRC 12693 | ND |
| 4 | *Rhodococcus qingshengii* dj1-6-2 | 10 ± 1.4 |
| 5 | *Staphylococcus capitis* JCM 2420 | ND |
| 6 | *Staphylococcus epidermidis* NBRC 100911 | ND |
| 7 | *Deinococcus radiodurans* R1 | 12 ± 4.2 |

ND: Not detected.

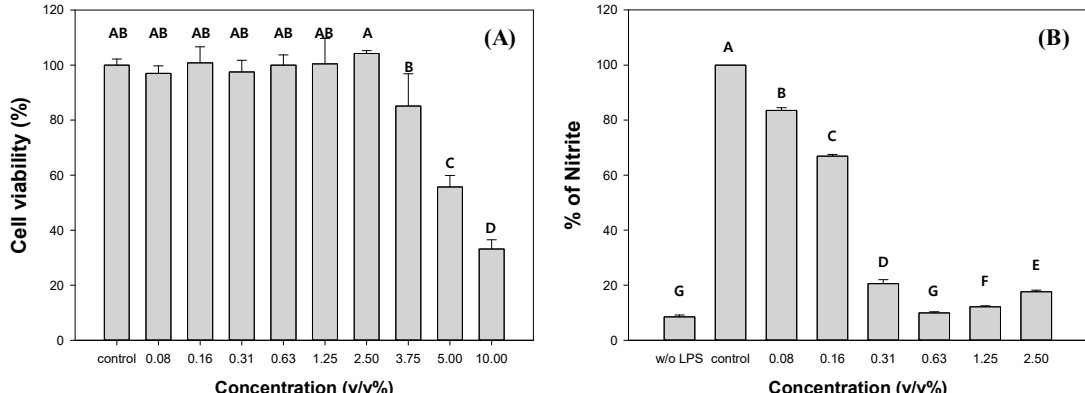

**Figure 2.** Cytotoxic (**A**) and anti-inflammatory (**B**) effects of *C. myrrha* (T.Nees) Engl. resin extracts in methanol (*w/o* LPS: no lipopolysaccharide and control: 0% of *C. myrrha* resin extracts). Conditions: RAW 264.7 cell with *C. myrrha* (T.Nees) Engl. extract at a concentration of 0.08–10% (*v/v*) for 48 h at 37 °C in a $CO_2$ incubator; NR assay (**A**); and NO production (**B**). The experiment was conducted in triplicate and the error bars represent the 95% confidence interval. AB, A, B, C, and D in (**A**) and A, B, C, D, E, F, and G in (**B**) indicate group classified from Tukey's test. The same letter means that there is no significant difference between the data.

*3.5. Evaluation of Antiviral Activity of C. myrrha (T.Nees) Engl. Resin Extracts*

　　The antiviral activity of methanol as the control and *C. myrrha* (T.Nees) Engl. resin extracted using methanol was evaluated against the H1N1 influenza virus. Methanol, *A. bidentata* root extract, and *C. myrrha* (T.Nees) Engl. resin extract showed 2.8%, 24.5%, and 81.2% virus RNA inhibition (from qRT-PCR results), respectively (Figure 3). These results imply that methanol did not inhibit viral RNA. In contrast, *C. myrrha* (T.Nees) Engl. resin extracts prepared using methanol can be associated with higher viral RNA inhibition due to the antiviral compounds of the extract.

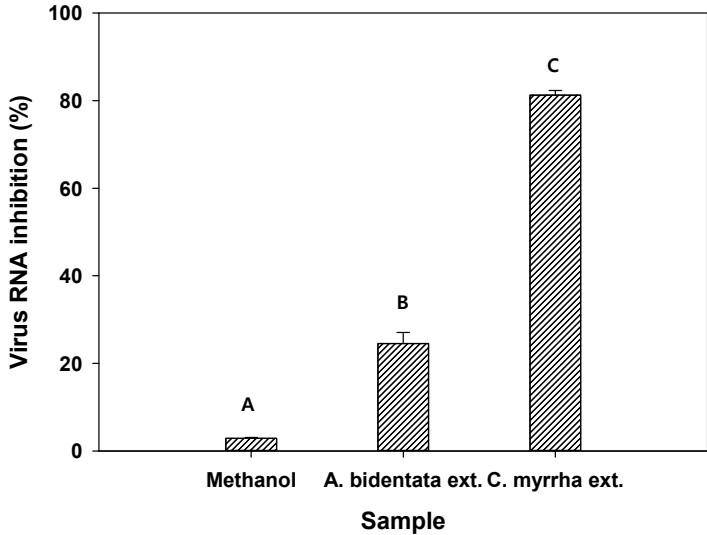

**Figure 3.** Antiviral activity of methanol as a control, *A. bidentata* Blume root extract as a positive control, and *C. myrrha* (T.Nees) Engl. resin extract using methanol against the H1N1 influenza virus. Conditions: H1N1 influenza virus (0.1 of MOI) in 1.5 mL of aqueous solution with 100 μL of methanol and *C. myrrha* (T.Nees) Engl. resin extracts in methanol; qRT-PCR. The experiment was conducted in triplicate and the error bars represent the 95% confidence interval. A, B, and C indicate group classified from Tukey's test. The same letter means that there is no significant difference between the data.

Based on the viral RNA inhibition results of the *C. myrrha* (T.Nees) Engl. resin extracts in methanol, RH-BC and WD-BC were coated with the *C. myrrha* (T.Nees) Engl. resin extracts to determine their antiviral activity. The remaining H1N1 influenza virus had decreased by about 22.6% and 24.3% in aqueous solutions with the *C. myrrha* (T.Nees) Engl. resin extracts coated with RH-BC and WD-BC compared with the initial virus containing aqueous solution (Table 4). In particular, the remaining viruses (96.5% and 96.7) after the reaction with RH-BC and WD-BC (without the *C. myrrha* (T.Nees) Engl. resin extract coating) were higher than those (77.4% and 75.6) after the reaction with *C. myrrha* (T.Nees) Engl. resin extracts coated with RH-BC- and WD-BC (Table 4). The difference in inhibitory effects also appeared according to the plaque assay (Figure S2). Therefore, *C. myrrha* (T.Nees) Engl. resin extracts bound to the surfaces of RH-BC and WD-BC may exhibit antiviral activity. These results suggest that there are some antiviral compounds (e.g., terpenoids) in *C. myrrha* resin extracts and that some antiviral compounds can be successfully coated onto biochar. For instance, curzerene, a type of terpenoid found in *C. myrrha* oil, is a possible antiviral compound [16]. Similarly, furanoeudesma-1,3-diene and curzerene, the major terpenoids, were observed in the present study, as reported by Madia et al. (Figure S3) [16]. The detailed quantities of furanoeudesma-1,3-diene and curzerene are described in Section 3.6.

**Table 4.** The remaining amount of virus in water (without virus), initial (virus in water), RH-BC and WD-BC without *C. myrrha* (T.Nees) Engl. resin extracts (final after reaction), and *C. myrrha* (T.Nees) Engl. resin extract-coated RH-BC and WD-BC (final after reaction) after the reaction. Conditions: H1N1 influenza virus (0.1 of MOI) in 3 mL of aqueous solution with 5 mg of *C. myrrha* (T.Nees) Engl. resin extract-coated biochar; shaken at 120 rpm and stored at 25 °C for 48 h.

| Sample | Ct Value | Remaining Amount of Virus (%) |
| --- | --- | --- |
| Water (without virus) | 0 | - |
| Virus (Initial) | 24.809 | 100.00 |
| WD-BC (Final) | 25.121 | 96.69 |
| Extract-coated WD-BC (Final) | 27.105 | 75.66 |
| RH-BC (Final) | 25.137 | 96.52 |
| Extract-coated RH-BC (Final) | 26.941 | 77.40 |

*3.6. Potential Mechanisms for Antibacterial and Antiviral Activity of C. myrrha (T.Nees) Engl. Resin Extracts*

Various bioactive compounds with antibacterial and antiviral activities need to be investigated because of the useful phytochemicals (e.g., polyphenols and terpenoids) present in plants and/or plant-based extracts. According to the results of HPLC and GC-MS analysis, polyphenols (i.e., tannic acid, rutin, quercetin, ferulic acid, caffeic acid, naringenin, and ellagic acid) and terpenoids (i.e., furanoeudesma-1,3-diene, curzerene, lindestrene, gazaniolide, -elemene, and -elemene) were detected in the *C. myrrha* (T.Nees) Engl. resin extracts in this study (Figure 1, Figure 4 and Figure S3).

As shown in Figure 1, tannic acid, rutin, and quercetin were the major polyphenols (above 20 g/mL) in the *C. myrrha* (T.Nees) Engl. resin extracts. As mentioned, the three major polyphenols (tannic acid, rutin, and quercetin) had significant antibacterial effects. In particular, tannic acid may lead to improved antibacterial activity by disturbing the uptake of sugars and amino acids, restricting bacterial growth. According to Wang et al., tannic acid inhibits the growth of *Clostridioides difficile* strains at concentrations above 16 g/mL [61]. Gullon et al., and Qi et al., reported that rutin and quercetin possess antimicrobial efficacy [62,63]. Polyphenols have been proposed as roles in the disruption of bacterial cell wall homeostasis, nucleic acid synthesis, and energy metabolism.

As shown in Figure S3, various terpenoids such as furanoeudesma-1,3-diene, curzerene, lindestrene, gazaniolide, β-elemene, γ-elemene, etc., were observed. Based on the results from previous literature regarding *C. myrrha* (T.Nees) Engl. resin extracts, furanoeudesma-1,3-diene and curzerene were measured as major compounds [16,64]. According to the

antiviral activity test, the viral RNA inhibition effects (81.2%) of *C. myrrha* (T.Nees) Engl. resin extracted using methanol were revealed (Figure 3). In contrast, there was negligible effect on virus RNA inhibition (2.8%) using methanol. The quantities of furanoeudesma-1,3-diene and curzerene in the terpenoids used in the antiviral activity experiment were 74.5 µg/mL and 12.4 µg/mL. Consequently, this finding may originate from terpenoids (i.e., furanoeudesma-1,3-diene and curzerene as the major compounds). The mechanisms of these terpenoids for antiviral activity were deduced to involve the interference of factors on the plasma membrane, viral attachment, antioxidant activity, and cell penetration [16].

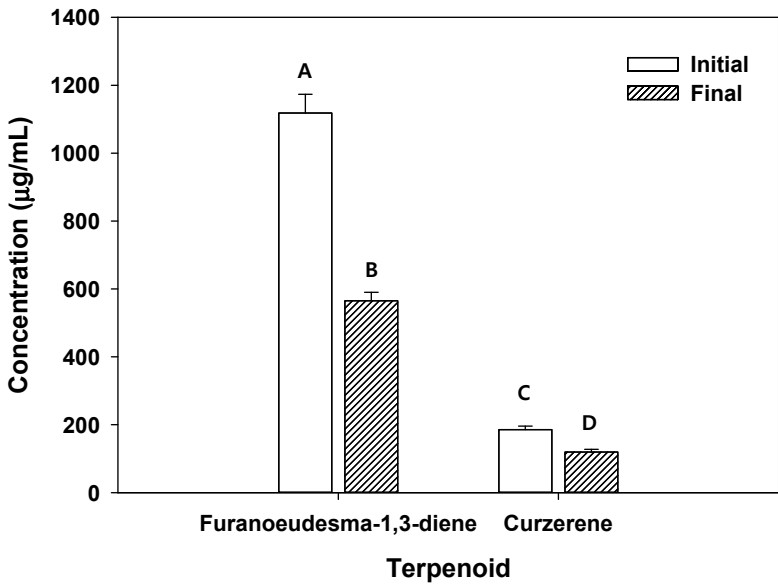

**Figure 4.** Concentration of terpenoids (i.e., furanoeudesma-1,3-diene and curzerene) in solution before and after adsorption of *C. myrrha* (T.Nees) Engl. resin extracts onto RH-BC. Conditions: 50 mg of biochar was mixed with 1 mL of the extract solution in a shaking incubator at 25 °C for 24 h at 120 rpm and dried at 60 °C for 24 h. The experiment was conducted in triplicate and the error bars represent the 95% confidence interval. A, B, C and D indicate group classified from Tukey's test. The same letter means that there is no significant difference between the data.

Additionally, furanoeudesma-1,3-diene (1118.42 µg/mL) and curzerene (185.34 µg/mL) were found in *C. myrrha* (T.Nees) Engl. resin extracted using methanol (Figure 4). After the precipitation of RH-BC in *C. myrrha* (T.Nees) Engl. resin extracts, the concentration of these terpenoids (e.g., furanoeudesma-1,3-diene and curzerene) decreased by about 49% (553.36 µg/mL) and 36% (66.89 µg/mL), respectively (Figure 4). These results suggest that these terpenoids were bound to the surface of RH-BC. The remaining H1N1 influenza virus decreased by approximately 22.6% in aqueous solution with the *C. myrrha* (T.Nees) Engl. resin extracts coated with RH-BC (Table 4). Possibly, this is due to the coated quantity of terpenoids (i.e., furanoeudesma-1,3-diene (18.4 µg/mL) and curzerene (2.2 µg/mL) onto RH-BC).

As illustrated in Figure 5, over 50% of the virus RNA inhibition was revealed in 50 µg/mL of furanoeudesma-1,3-diene and 70 µg/mL of curzerene. Low concentrations of furanoeudesma-1,3-diene (2.5–30 µg/mL) and curzerene (over 60 µg/mL) only affect up to 50% of virus inhibition [16]. The results of the present study are consistent with those of the previous studies [16]. The effects of viral RNA inhibition in the aqueous solution with the *C. myrrha* (T.Nees) Engl. resin extracts coated with biochar were lower than those in only *C. myrrha* (T.Nees) Engl. resin extracts. This may be attributed to the low binding concentration of terpenoids into biochar and the lower concentration of terpenoids compared with only *C. myrrha* (T.Nees) Engl. resin extracts. These effects of viral RNA inhibition through biochar can be easily solved by increasing the amount of biochar added and enhancing terpenoid quantity binding with the activation and functionalization of

biochar. Therefore, *C. myrrha* (T.Nees) Engl. resin extract-coated biochar has the potential to be used in various applications such as cosmetics, air filtration, fertilizers, drug delivery, and corrosion inhibition.

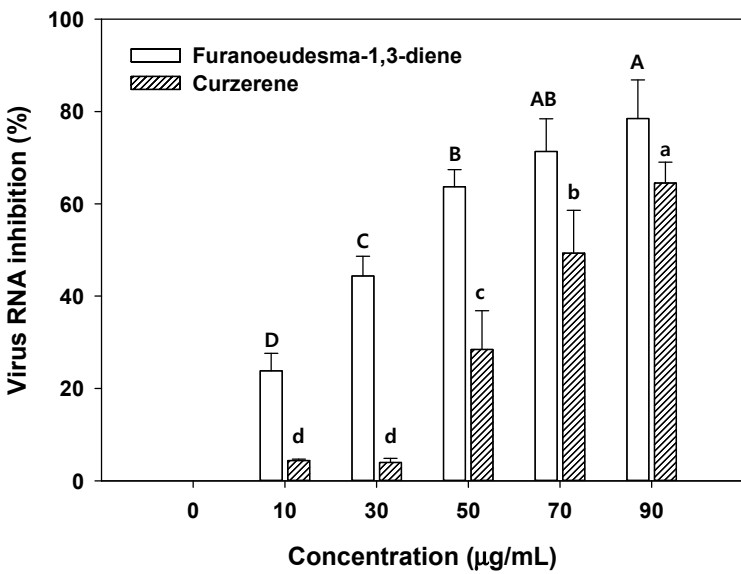

**Figure 5.** Antiviral activity of concentration dependence of terpenoids (furanoeudesma-1,3-diene and curzerene) against the H1N1 influenza virus. Conditions: H1N1 influenza virus (0.1 of MOI) in 1.5 mL of aqueous solution ranging from 10–90 μL/mL. The experiment was conducted in triplicate and the error bars represent the 95% confidence interval. AB, A, B, C, and D and a, b, c, and d indicate group classified from Tukey's test. The same letter means that there is no significant difference between the data.

In addition, *C. myrrha* (T.Nees) Engl. resin extract-coated biochars (e.g., RH-BC and WD-BC) with different properties (feedstock (e.g., RH-BC; rice husk and WD-BC; and wood powder), BET surface area (i.e., RH-BC; 205.6 m$^2$/g and WD-BC; and 211.1 m$^2$/g), and functional groups on the surface) were evaluated for their antiviral activity (Table S1 and Figure S4). The Brunauer–Emmett–Teller (BET) surface areas of RH-BC and WD-BC were similar; however, their functional groups were different (Figure S4). Accordingly, we deduced the differences in the adsorption capacity of terpenoids onto the biochar. To confirm terpenoid binding, changes in the functional groups on the surface of the RH-BC were analyzed before and after precipitation (e.g., adsorption) using FTIR. As shown in Figure 6, 875 cm$^{-1}$, 1049 cm$^{-1}$, 1375 cm$^{-1}$, 1430 cm$^{-1}$, 1740 cm$^{-1}$, 2930 cm$^{-1}$, and 3200–3400 cm$^{-1}$ peaks, indicating $CO_3^{2-}$-, C-O stretching, C=O carboxylate ion stretching, C-H bending, C=O group, -CH group, and OH group in RH-BC and *C. myrrha* resin extract-coated RH-BC appeared [35,65]. Among them, the peaks at 875 cm$^{-1}$ and 1049 cm$^{-1}$, indicating $CO_3^{2-}$- and C-O stretching, had decreased and the peak at 1740 cm$^{-1}$ indicated an increase in the C=O group. Hence, these results (e.g., the increase in the C=O group) imply that the terpenoids, such as furanoeudesma-1,3-diene and curzerene, bind with the functional groups (e.g., $CO_3^{2-}$ and C-O stretching) on the surface of RH-BC or the π-π interaction on the surface area.

As shown in the XPS results, C1s, O1s, Si2s, and Si2p peaks appeared in the survey scans of the RH-BC and *C. myrrha* (T.Nees) Engl. resin extract-coated RH-BC (Figure 7). In the comparison between RH-BC and *C. myrrha* (T.Nees) Engl. resin extract-coated RH-BC, the C1s, O1s, Si2s, and Si2p peaks decreased after coating, indicating that various compounds in the *C. myrrha* (T.Nees) Engl. resin extracts were bound to the surface of RH-BC. These peak intensities (e.g., C1s, O1s, Si2s, and Si2p) were similar at 25, 50, and 100 °C, although they slightly decreased at 200 °C. These phenomena can be attributed to the volatility of various compounds in the *C. myrrha* resin extracts at high temperatures (200 °C).

Nevertheless, the C1s, O1s, Si2s, and Si2p peaks were still present. These results demonstrate that various compounds in *C. myrrha* (T.Nees) Engl. resin extracts are strongly incorporated onto the surface of the biochar. Therefore, biochar is a promising material for reducing the loss of desired compounds. In addition, *C. myrrha* (T.Nees) Engl. resin extract-coated RH-BC can be applied as a potential material for antibacterial and antiviral activity via the successful binding of phytochemicals (e.g., terpenoids) to the surface of the biochar. Further evaluation of the binding correlation between the properties of biochar and terpenoids, as well as more detailed binding mechanisms, are necessary.

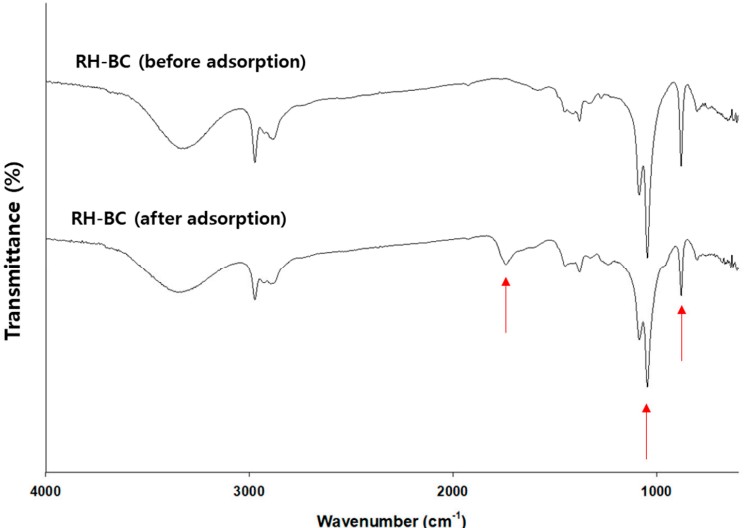

**Figure 6.** FTIR spectra of the RH-BCs before and after *C. myrrha* (T.Nees) Engl. resin extract adsorption. The arrow means the changes (i.e., increase or decrease) of peaks after adsorption.

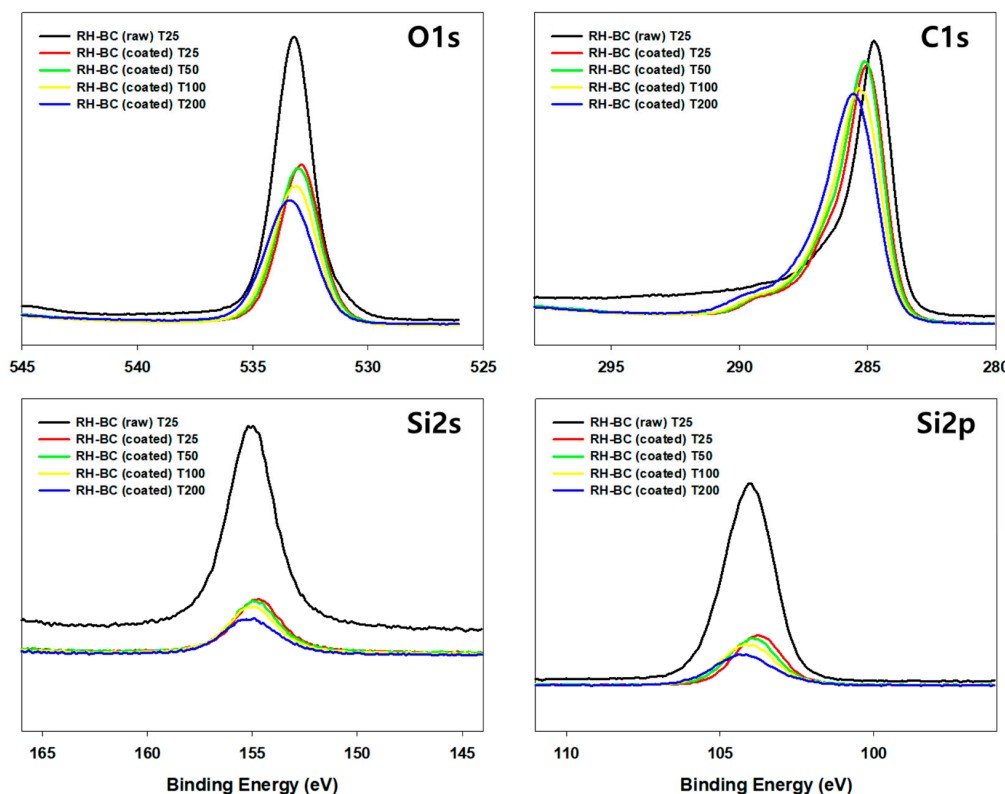

**Figure 7.** XPS spectra of raw RH-BC (not coated) and *C. myrrha* (T.Nees) Engl. resin extract-coated RH-BC (coated) at various temperatures (25 °C, 50 °C, 100 °C, and 200 °C) for adsorption stability evaluation.

## 4. Conclusions and Future Research

To extract effective compounds from *C. myrrha* (T.Nees) Engl. resin, various solvents with different polarities and dielectric constants were used as extraction solvents. Among the *C. myrrha* (T.Nees) Engl. resin extracts prepared, the methanolic extract showed significant antibacterial activity against Gram-positive bacteria. This may be due to the distinct solvent properties, which lead to the extraction of different effective compounds. Particularly, the extract prepared by soaking in methanol exhibited anti-inflammatory activity at non-cytotoxic concentrations. According to the results of the component analysis using HPLC and GC-MS, the extracts contained various polyphenols (e.g., tannic acid, rutin, and quercetin) and terpenoids (e.g., furanoeudesma-1,3-diene and curzerene), which are known to have antibacterial, anti-inflammatory, and antiviral effects. Furthermore, novel biochar-based materials used to introduce the functionality of *C. myrrha* (T.Nees) Engl. resin extracts for the investigation of antiviral activity were fabricated using a simple adsorption process. The *C. myrrha* (T.Nees) Engl. resin extract-coated biochars, which adsorbed terpenoids, showed antiviral activity against the H1N1 influenza virus, along with extracts that existed in the liquid state. Therefore, biochar coated with methanolic *C. myrrha* (T.Nees) Engl. resin extracts suggest the possibility of using natural-derived substances in various fields (e.g., purification, agriculture, pharmaceuticals, cosmetics, and food packaging) as they are fabricated as novel materials via the adsorption of effective compounds.

In future research, there are many interesting issues that should be considered. It is worth studying the relationship between antiviral compounds and viruses. Firstly, *C. myrrha* (T.Nees) Engl. resin extracts for antiviral activity against the H1N1 influenza virus will be evaluated in more detail including CPE (cytopathic effect), Western blotting, and so on. Also, the comparison of antiviral agents and *C. myrrha* (T.Nees) Engl. resin extracts for antiviral activity against the H1N1 influenza virus and the evaluation of *C. myrrha* (T.Nees) Engl. resin extracts for antiviral activity against other viruses will be applied. In addition, exploring the application of biochar is a very challenging topic. Hence, the adsorption capacity of the antibacterial and antiviral compounds in *C. myrrha* (T.Nees) Engl. resin extracts onto various biochars (rice husk, wood, grass, sludge, microalgae, etc., -based biochars) will be explored in more detail. To evaluate the adsorption of compounds onto biochar, the kinetic and isotherm studies will be analyzed.

**Supplementary Materials:** The following supporting information can be downloaded at: https://www.mdpi.com/article/10.3390/app131810549/s1, Table S1: Brunauer–Emmett–Teller (BET) surface areas of the RH-BC and WD-BC; Figure S1: HPLC spectra of authentic polyphenols for polyphenol analysis; Figure S2: Inhibitory effects of RH-BC (A) without *C. myrrha* (T.Nees) Engl. resin extracts and RH-BC (B) coated with *C. myrrha* (T.Nees) Engl. resin extracts against the H1N1 influenza virus using plaque assay; Figure S3: GC-MS spectra of *C. myrrha* (T.Nees) Engl. resin extracts for terpenoid analysis; Figure S4: FTIR spectra of RH-BC and WD-BC before *C. myrrha* (T.Nees) Engl. resin extract adsorption.

**Author Contributions:** J.W.K., methodology, formal analysis, writing—original draft preparation; S.P., formal analysis, writing—original draft preparation; Y.W.S., resources, data curation; H.J.S., methodology, data curation; S.W.Y., methodology; J.H., data curation; J.W.J., formal analysis; I.-S.L., data curation; S.H.L., writing—review and editing; Y.-K.C., conceptualization, writing—original draft preparation, supervision; and H.J.K., conceptualization, writing—original draft preparation, supervision. All authors have read and agreed to the published version of the manuscript.

**Funding:** This research was supported by the Korea Water Cluster as a 2023 Demand-based Carbon-neutrality Water-tech Demonstration Supporting Project (Project No. B0080612001292) and the Korea Forest Service as an R&D Program for Forest Science Technology (Project No. 2023431B10-2324-0802).

**Institutional Review Board Statement:** Not applicable.

**Informed Consent Statement:** Not applicable.

**Data Availability Statement:** Not applicable.

**Conflicts of Interest:** The authors declare no conflict of interest.

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
