# Peer review of "Evaluation of Antibacterial and Antiviral Compounds from Commiphora myrrha (T.Nees) Engl. Resin and Their Promising Application with Biochar"

_applsci, doi:10.3390/app131810549_

Round 1

Reviewer 1 Report

Despite the interesting topic, the manuscript needs to be really improved before being taken into consideration for publication.

1. In the title the authors mentioned just the antiviral activity, in the main text the antibacterial activity was described. 

2. At the first mention in title, abstract and main text, the botanical species must be named with the complete name: genus + species + author(s); for your convenience check the website https://www.worldfloraonline.org/

3. lines 12-13: the sentence is not clear. When the authors mention "different polarity and dielectric constant" what do the authors mean? Different than?

4. lines 15-16: the sentence "Furanoeudesma-1,3-diene and curzerene... were analyzed using GC-MS '' should be written in a clearer way. In fact, the whole extract was analyzed by GCMS and Furanoeudesma-1,3-diene and curzerene were found as the main compounds.

5. The introduction must be written again. A lot of obvious and well known information are given. I suggest the authors focus more on the topic of their work.

Some example of unnecessary information

-Natural compounds (e.g., phytochemicals), including flavonoids, phenolic compounds, terpenes, terpenoids..... including antioxidant, antimicrobial, antiviral, and anti-inflammatory [3-7].

-Extracts of Pimpinella species ..... anticancer effects, and with low cytotoxicity

(Just to give a couple of examples)

6. Par 2.2: more information about extract preparation are needed. Were the solvent evaporated? (according to lines 122-125 the extracts after filtration were used as they are); eventually, how did they evaporate DMSO? How is the yield of extraction?

7. How is the concentration of extracts used to screen the optimal solvent? (table 2)

8.  Parag 3.2:  the sentence "According to previous studies, the solvent polarity index and dielectric ..... and solubility of the compounds [20, 51-53]" is unnecessary (it is known that different solvents give different extract).

9. The HPLC chromatogram must be reported. How do they identify each polyphenol? It is strange to me that this kind of molecule was found in resin (more precisely in a gum oleoresin that Commiphora sp. produce). Usually there are polar components (gum that is made of carbohydrate) while the other part is made of nonpolar compounds (terpenes and terpenoids).

10. Figure 2A: It seems that increasing the concentration of extract the cell viability increases from 0.63 to 10, why?

11. Figure 2B: Could the authors explain in the legend what they mean for w/o LPS and control? 

12. To coat the biochar, the authors said "....was dried at 60 °C for 24 h to remove methanol.." Can the myrrh components be evaporated, in part? How were chemically characterized the coated-biocar?

Reviewer 2 Report

The given study by Jin Woo Kim and coauthors discusses the evaluation of antiviral compounds from Commiphora molmol myrrh resin and their potential application with biochar. The authors explore the antibacterial properties of myrrh resin extracts and identify the main terpenoids responsible for their effectiveness. They also discuss the potential use of biochar in conjunction with myrrh resin extracts for antiviral applications. It is a good work as natural bioactives are being exploited for beneficial use. However, certain flaws were discovered in this version that must be addressed before further consideration:

Comparing the results with at least one known in vitro antiviral or plant extract (methanol) of H1N1 influenza virus as reference compound should be done. And experimental data compared with a positive control should be included in the figures and they should be disused in the result section as well.  

what was the concentration and volume of methanol (solvent) in mock/control wells?

Authors should determine the maximum non-toxic dose of the extract/compound along with CC50, EC50 or IC50 before doing any inhibitory assay on RAW 264.7 cell line.

In antiviral assays, MOI of the virus should me mentioned instead of virus concentration (PFU/ml) and volume.

Authors should include microscopic images of CPE compared to mock control (with and without compounds/methanol and no virus) in supplementary information.

More experiments are needed to validate the antiviral activity of the extract by viral inhibition at protein level by western blotting, or by a gold standard-plaque inhibition assay.

Authors should re-read and check all the figure legends. A legend should have proper title, the materials & methods involved with the presented figure (VERY brief to describe the design of the experiment), any other miscellaneous details such as explaining abbreviations, dyes used/color or image scale. What were the sample / treatment groups? n=? and statistical analyses carried out.

Instead of Conclusion, it may be replaced with Conclusion and Future research. In that case, there should be some restructuring of the text with inclusion of some promising upcoming areas of futuristic research in this direction.

Round 2

Reviewer 1 Report

Most of the issues have not been solved.

1. The HPLC chromatogram that authors attached as supporting material is just the HPLC of polyphenols. In that way it is still not clear how they detected polyphenols in myrrh resin. 

2. The manuscript still contains unnecessary and obvious sentences. Just one example among others. "Extracts of Pimpinella species have been found to contain anethole, which has antimicrobial, anticancer, and so on [6].” Why do the authors mention Pimpinella species? Which is the connection with Commiphora?

3. The botanical nomenclature is not used properly in the whole manuscript.

4. The authors claimed that the extracts were used without evaporating the solvents. The solvents themselve have antibacterial/antiviral activity

Reviewer 2 Report

The authors justified some of the concerns and revised the work according.

Please add statistical significance/p values in the graphs.
